# On Solving Nonlinear Moving Boundary Problems with Heterogeneity Using the Collocation Meshless Method

**Cheng-Yu Ku** [1,2], **Jing-En Xiao** [1,*] **and Chih-Yu Liu** [1]

[1] Department of Harbor and River Engineering, National Taiwan Ocean University, Keelung 20224, Taiwan; chkst26@mail.ntou.edu.tw (C.-Y.K.); 20452003@email.ntou.edu.tw (C.-Y.L.)

[2] Center of Excellence for Ocean Engineering, National Taiwan Ocean University, Keelung 20224, Taiwan

[*] Correspondence: 20452002@email.ntou.edu.tw; Tel.: +886-2-2462-2192 (ext. 6159)

**Abstract:** In this article, a solution to nonlinear moving boundary problems in heterogeneous geological media using the meshless method is proposed. The free surface flow is a moving boundary problem governed by Laplace equation but has nonlinear boundary conditions. We adopt the collocation Trefftz method (CTM) to approximate the solution using Trefftz base functions, satisfying the Laplace equation. An iterative scheme in conjunction with the CTM for finding the phreatic line with over–specified nonlinear moving boundary conditions is developed. To deal with flow in the layered heterogeneous soil, the domain decomposition method is used so that the hydraulic conductivity in each subdomain can be different. The method proposed in this study is verified by several numerical examples. The results indicate the advantages of the collocation meshless method such as high accuracy and that only the surface of the problem domain needs to be discretized. Moreover, it is advantageous for solving nonlinear moving boundary problems with heterogeneity with extreme contrasts in the permeability coefficient.

**Keywords:** free surface; nonlinear; heterogeneity; the collocation Trefftz method; nonlinear boundary condition

## 1. Introduction

The free surface flow is a moving boundary problem governed by the Laplace equation but has nonlinear boundary conditions. The study of free surface seepage problem plays a crucial role in the analysis of hydraulic engineering. In the design of embankment, earth dams and rock–fill dams, finding the position of the moving boundary is of importance [1,2]. The solution of the Laplace governing equation may be carried out by solving the boundary value problem. Because the solution of the free surface seepage flow is nonlinear, iterative techniques are often required in the solution process for matching the over–specified boundary conditions. Various mesh–based numerical methods [3–13] have been used for the analysis of free surface flow. For the mesh–based methods, automatic grid regeneration [6] is commonly used to solve the free surface problems in mesh–based approaches. However, convergence problems are often raised due to the changing of element shapes and types in the process of the mesh generation. While the complexity of the boundary conditions is considered, mesh–based methods may become unstable since the automatic grid regeneration is likely to generate distorted meshes.

Recently, meshless methods have attracted much attention to solve free surface seepage problems [14]. Compared to mesh–based methods, the discretization of the domain for meshless methods is relatively simple because only arbitrary collocation points need to be placed in the physical

domain without using any elements. If the basic functions satisfy the governing equation, the collocation points may be conducted only on the boundary. Accordingly, the meshless method has advantages with problems involving complex geometry [15]. Among several meshless methods, the collocation Trefftz method (CTM) may be regarded as an attractive boundary–type meshless method [16]. In the past, study of the Trefftz method was less widespread because the ill-posedness of the CTM limits the applications of the method. However, using the characteristic length [17–19], the CTM has been adopted to obtain accurate solutions for solving the Laplace governing equation in three–dimensions enclosed by simply and multiply connected domains [20].

The Trefftz method is first proposed by the German mathematician Erich Trefftz. Later, the CTM [21–30] is commonly used for solving partial differential equations. Since the CTM is categorized into the boundary–type meshless method, it approximates the solutions of the governing equation using the Trefftz basis functions where the solutions are described as the assembly of the Trefftz functions [31]. The CTM requires the evaluation of the coefficients in which they may be obtained by solving the linear simultaneous equations assembled by using the boundary conditions at a number of collocation points. Applications of the CTM such as Laplace and modified Helmholtz equations [32,33] and the problem of boundary detection [34] has been studied. Due to the complexity, applications of the CTM are most limited to the homogeneous problems. In addition, the study of nonlinear moving boundary problems with heterogeneity using the CTM has not been reported yet.

This paper presents the study on solving nonlinear moving boundary problems in heterogeneous geological media using the CTM. The free surface flow is a moving boundary problem governed by Laplace equation but has nonlinear boundary conditions. We adopt the CTM to approximate the solution using Trefftz base functions satisfying the Laplace equation. An iterative scheme in conjunction with the CTM for finding the phreatic line with over–specified nonlinear moving boundary conditions is developed. To deal with flow in the layered heterogeneous soil, the domain decomposition method is used so that the hydraulic conductivity in each subdomain can be different. The method proposed in this study is verified by several numerical examples. The formulation of the proposed method is described as follows.

## 2. Governing Equation and Boundary Conditions

The two-dimensional Laplace equation used to represent flow through a homogenous rectangular dam is expressed as

$$\Delta\varphi = 0 \text{ in } \Omega \tag{1}$$

and

$$\varphi = g \text{ on } \Gamma_D, \tag{2}$$

$$\varphi_n = f \text{ on } \Gamma_N, \tag{3}$$

where $\varphi$ is the head, $\Delta$ is the Laplacian, $\Omega$ represents the domain boundary of the problem, $g$ and $f$ denote the Dirichlet and Neumann boundary conditions, respectively. $n$ denotes the normal vector. $\Gamma_D$ and $\Gamma_N$ denote the Dirichlet and Neumann boundary conditions.

The boundary conditions of the rectangular dam with a moving boundary, as depicted in Figure 1a, can be presented by $\Gamma_1, \Gamma_2, \Gamma_3, \Gamma_4$ and $\Gamma_5$. The Dirichlet boundary conditions are imposed on the $\Gamma_2$ and $\Gamma_5$, respectively.

$$\varphi = H_2 \text{ on } \Gamma_2, \tag{4}$$

$$\varphi = H_1 \text{ on } \Gamma_5, \tag{5}$$

where $H_2$ denotes the downstream elevation, $H_1$ denotes the upstream elevation. Neglecting the velocity head, the head is expressed as

$$\varphi = Y(x) + \frac{p}{\gamma}, \tag{6}$$

where $Y(x)$ denotes the height above the sea level, $p$ denotes the pore water pressure and $\gamma$ denotes the water unit weight. On $\Gamma_4$, the over–specified moving boundary conditions are given as

$$\frac{\partial \varphi}{\partial n} = 0, \ \varphi = Y(x) \text{ on } \Gamma_4. \tag{7}$$

On $\Gamma_3$, the seepage face boundary condition is depicted as

$$\varphi = Y(x) \text{ on } \Gamma_3. \tag{8}$$

$\varphi = Y(x)$ is unknown which can be solved by using the iterative scheme. On $\Gamma_1$, the no–flow condition is given as

$$\frac{\partial \varphi}{\partial n} = 0 \text{ on } \Gamma_1. \tag{9}$$

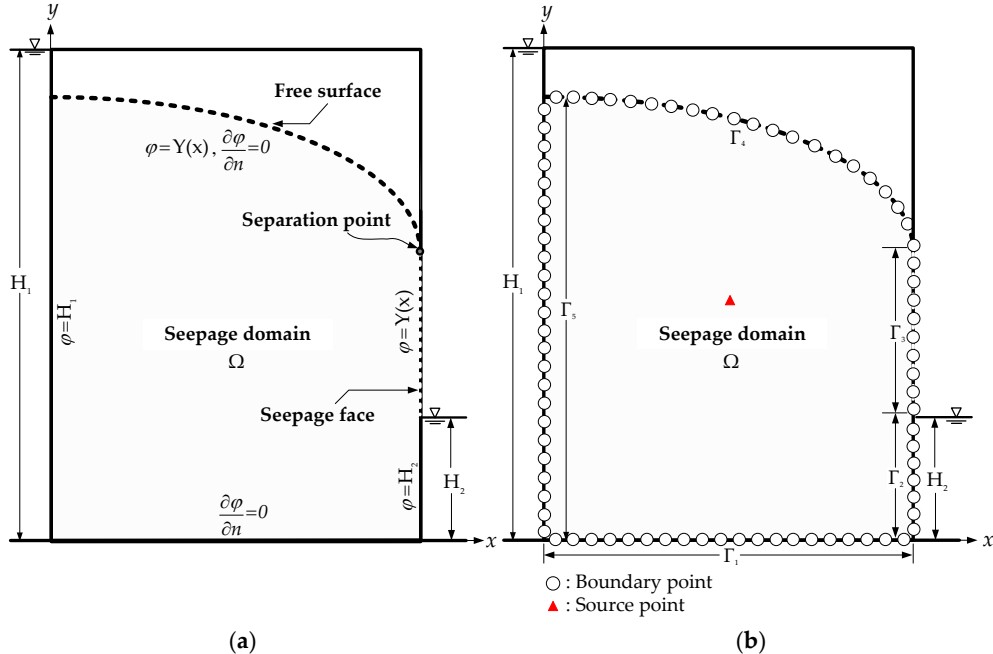

**Figure 1.** Nonlinear moving surface through a rectangular dam. (**a**) The cross section and boundary conditions and (**b**) collocation points on the boundary.

## 3. The Collocation Trefftz Method

The Laplace equation in polar coordinate system is expressed as

$$\frac{\partial^2 \varphi}{\partial r^2} + \frac{1}{r}\frac{\partial \varphi}{\partial r} + \frac{1}{r^2}\frac{\partial^2 \varphi}{\partial \theta^2} = 0, \tag{10}$$

where the radial coordinate is denoted by $r$ and the angular coordinate is denoted by $\theta$. The solution of the Laplace governing equation is approximated by using the Trefftz basis functions satisfying the governing equation, as shown in Equation (10). The Trefftz basis functions are obtained by finding the general solutions using the separation of variables method [35]. The Trefftz basis functions can be found to solve problems in a simply connected domain, as shown in Figure 2.

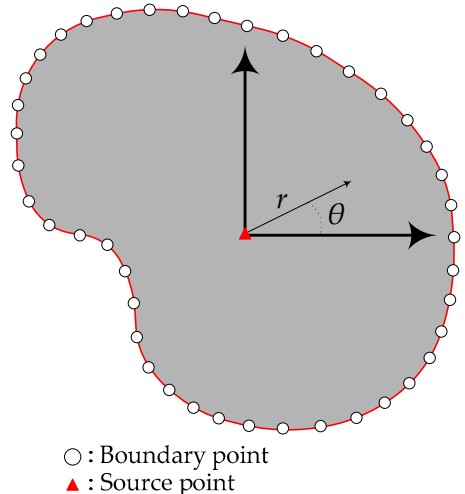

○ : Boundary point
▲ : Source point

**Figure 2.** Illustration of the collocation scheme in the CTM.

### 3.1. Formulation of T-Complete Basis Functions

We may apply the separation of variables [36]. The solution may be in the following form:

$$\varphi(r,\theta) = U(r)V(\theta). \tag{11}$$

For simplicity, we let

$$U' = \frac{dU(r)}{dr}, \; U'' = \frac{d^2U(r)}{dr^2} \text{ and } V'' = \frac{d^2V(\theta)}{d\theta^2}. \tag{12}$$

Then, Equation (10) can be rewritten as follows.

$$U''V + \frac{1}{r}U'V + \frac{1}{r^2}UV'' = 0. \tag{13}$$

We divide $U(r)V(\theta)$ on both sides in the above equation and the equation can be rewritten as two differential equations as follows.

$$r^2\frac{U''}{U} + r\frac{U'}{U} = \lambda r^2, \tag{14}$$

$$\frac{V''}{V} = -\lambda. \tag{15}$$

Using the constant $v$ to ensure positive or negative constants, we have $\lambda = 0$, $\lambda = v^2$ and $\lambda = -v^2$. Considering the first scenario $\lambda = 0$, we obtain the solutions as follows.

$$V = D_1\theta + D_2, \tag{16}$$

$$U = D_3 \ln r + D_4, \tag{17}$$

where $D_1$, $D_2$, $D_3$ and $D_4$ are constants. Using the boundary conditions of $V(r,0) = V(r,2\pi)$, we may find that $D_1 = 0$. Substituting Equations (16) and (17) into Equation (11), we have

$$\varphi = a_0 + b_0 \ln r, \tag{18}$$

where $a_0$ and $b_0$ denote the coefficients. Considering the second scenario, $\lambda = v^2$, we obtain the following solutions.

$$V = D_5 \cos(v\theta) + D_6 \sin(v\theta), \tag{19}$$

$$U = D_7 r^v + D_8 r^{-v}, \tag{20}$$

where $D_5$, $D_6$, $D_7$ and $D_8$ are the coefficients. Inserting the above equations into Equation (11), we obtain

$$\varphi = a r^v \cos(v\theta) + b r^v \sin(v\theta) + c r^{-v} \cos(v\theta) + d r^{-v} \sin(v\theta), \tag{21}$$

where $a$, $b$, $c$ and $d$ denote the coefficients. Then, we may consider the last scenario $\lambda = -v^2$. Since there is not able to find any non–zero periodic solutions of differential system for $U(r)$, we may only find $V(\theta) = 0$. Collecting all the solutions from the above results, the linearly independent solutions to Laplace equation can be obtained as follows.

$$\left\{ 1, \quad \ln r, \quad r^v \cos(v\theta), \quad r^v \sin(v\theta), \quad r^{-v} \cos(v\theta), \quad r^{-v} \sin(v\theta) \right\}. \tag{22}$$

The Trefftz basis functions in a simply connected domain are as follows.

$$\left\{ 1 \quad r^v \cos(v\theta), \quad r^v \sin(v\theta) \right\}. \tag{23}$$

In the numerical analysis, we approach the general solution in the form of infinite series of the Laplace equation in a simply connected domain by using a finite number of $m$. As a result, Equation (23) can be rewritten as

$$\varphi = a_0 + \sum_{v=1}^{m} \{a_v r^v \cos(v\theta) + c_v r^v \sin(v\theta)\}, \tag{24}$$

where $m$ represents the terms of the Trefftz order. The above Equation (24) can be used to match the Dirichlet boundary condition. We may also need to consider the Neumann boundary conditions as follows.

$$\varphi_n = \frac{\partial \varphi}{\partial n} \text{ on } \Gamma_N. \tag{25}$$

Equation (25) can be rewritten as

$$\frac{\partial \varphi}{\partial n} = \nabla \varphi \cdot \vec{n}, \tag{26}$$

where $\nabla$ is the gradient and $\vec{n} = (n_x, n_y)$ denotes the normal vector. Equation (25) can then be written as

$$\frac{\partial \varphi}{\partial n} = \frac{\partial \varphi}{\partial x} n_x + \frac{\partial \varphi}{\partial y} n_y, \tag{27}$$

where

$$\frac{\partial \varphi}{\partial x} = \frac{\partial \varphi}{\partial r} \frac{\partial r}{\partial x} + \frac{\partial \varphi}{\partial \theta} \frac{\partial \theta}{\partial x}, \quad \frac{\partial \varphi}{\partial y} = \frac{\partial \varphi}{\partial r} \frac{\partial r}{\partial y} + \frac{\partial \varphi}{\partial \theta} \frac{\partial \theta}{\partial y}. \tag{28}$$

Using Equation (24), we may find the derivatives of $\partial \varphi / \partial r$ and $\partial \varphi / \partial \theta$ as follows.

$$\frac{\partial \varphi}{\partial r} = \sum_{v=1}^{m} a_v v r^{v-1} \cos(v\theta) + c_v v r^{v-1} \sin(v\theta), \tag{29}$$

$$\frac{\partial \varphi}{\partial \theta} = \sum_{v=1}^{m} a_v (-v) r^v \sin(v\theta) + c_v v r^v \cos(v\theta). \tag{30}$$

Using Equations (29) and (30), Equation (27) leads to

$$\begin{aligned}
\frac{\partial \varphi}{\partial n} = \sum_{v=1}^{m} &\{a_v [(v r^{v-1} \cos(v\theta) \cos\theta + (-v) r^v \sin(v\theta) \tfrac{-\sin\theta}{r}) n_x \\
&+ (v r^{v-1} \cos(v\theta) \sin\theta + (-v) r^v \sin(v\theta) \tfrac{\cos\theta}{r}) n_y] \\
&+ c_v [(v r^{v-1} \sin(v\theta) \cos\theta + v r^v \cos(v\theta) \tfrac{-\sin\theta}{r}) n_x \\
&+ (v r^{v-1} \sin(v\theta) \sin\theta + v r^v \cos(v\theta) \tfrac{\cos\theta}{r}) n_y] \}
\end{aligned} \tag{31}$$

### 3.2. The Characteristic Length

The characteristic length plays a crucial role in controlling the proposed numerical approach in a stable way. Because the matrix assembled with Trefftz trial functions is a full matrix, the resultant system of linear equations may be ill–posed [17,18]. The accuracy of the results from the Trefftz method depends sensitively on the order of the T-complete basis functions. Besides, the numerical solution may be unstable. Related to the CTM for solving two-dimensional Laplacian problems, Liu [17] proposed a characteristic length to mitigate the problems of the ill-posedness for the system of linear equations. Applying Dirichlet boundary condition, we obtain

$$\varphi = a_0 + \sum_{v=1}^{m} [a_v (\frac{r}{R})^v \cos(v\theta) + c_v (\frac{r}{R})^v \sin(v\theta)]. \tag{32}$$

Using the CTM, we obtain the approximation solution of the Laplace equation as follows.

$$\varphi(\mathbf{x}) = \sum_{v=1}^{m} \mathbf{b}_v \mathbf{T}_v(\mathbf{x}), \tag{33}$$

where $\mathbf{b}_v = [\begin{array}{ccc} a_0 & a_v & c_v \end{array}]$, $\mathbf{T}_v = [\begin{array}{ccc} 1 & (r/R)^v \cos(v\theta) & (r/R)^v \sin(v\theta) \end{array}]$, $\mathbf{x}$ is the coordinate of the collocation points and $\mathbf{x} \in \Omega$. Applying the Neumann boundary condition, we may obtain the following equations for simply connected domain using the characteristic length.

$$\begin{aligned}
\frac{\partial \varphi}{\partial n} = \sum_{v=1}^{m} \{ & a_v [(v(\tfrac{1}{R})^v r^{v-1} \cos(v\theta) \cos\theta + (-v)(\tfrac{1}{R})^v r^v \sin(v\theta)\tfrac{-\sin\theta}{r})n_x \\
& + (v(\tfrac{1}{R})^v r^{v-1} \cos(v\theta) \sin\theta + (-v)(\tfrac{1}{R})^v r^v \sin(v\theta)\tfrac{\cos\theta}{r})n_y] \\
& + c_v [(v(\tfrac{1}{R})^v r^{v-1} \sin(v\theta) \cos\theta + v(\tfrac{1}{R})^v r^v \cos(v\theta)\tfrac{-\sin\theta}{r})n_x \\
& + (v(\tfrac{1}{R})^v r^{v-1} \sin(v\theta) \sin\theta + v(\tfrac{1}{R})^v r^v \cos(v\theta)\tfrac{\cos\theta}{r})n_y] \}
\end{aligned} \tag{34}$$

To mitigate the ill-posedness, the characteristic length [19], $R$, is adopted and is expressed as

$$R = 1.5 \times maximum(r), \tag{35}$$

where $maximum(r)$ denotes the maximum radial distance in the problem domain. After adopting the characteristic length in our numerical model, the ill-posed phenomenon is greatly reduced, and the accurate numerical solutions can be obtained. Collocating the numerical expansion from Equations (32) and (34) at boundary collocation points to match the given boundary conditions, we may obtain the following equation.

$$\mathbf{Tb} = \mathbf{B}, \tag{36}$$

$$\mathbf{T} = \begin{bmatrix} 1 & (r_1/R)\cos(\theta_1) & (r_1/R)\sin(\theta_1) & \cdots & (r_1/R)^m \cos(m\theta_1) & (r_1/R)^m \sin(m\theta_1) \\ 1 & (r_2/R)\cos(\theta_2) & (r_2/R)\sin(\theta_2) & \cdots & (r_2/R)^m \cos(m\theta_2) & (r_2/R)^m \sin(m\theta_2) \\ \vdots & \vdots & \vdots & \cdots & \vdots & \vdots \\ 1 & (r_i/R)\cos(\theta_i) & (r_i/R)\sin(\theta_i) & \cdots & (r_i/R)^m \cos(m\theta_i) & (r_i/R)^m \sin(m\theta_i) \\ 0 & N^{a_1}_{1,v=1} & N^{c_1}_{1,v=1} & \cdots & N^{a_v}_{1,v=m} & N^{c_v}_{1,v=m} \\ 0 & N^{a_1}_{2,v=1} & N^{c_1}_{2,v=1} & \cdots & N^{a_v}_{2,v=m} & N^{c_v}_{2,v=m} \\ \vdots & \vdots & \vdots & \cdots & \vdots & \vdots \\ 0 & N^{a_1}_{j,v=1} & N^{c_1}_{j,v=1} & \cdots & N^{a_v}_{j,v=m} & N^{c_v}_{j,v=m} \end{bmatrix}, \tag{37}$$

$$\mathbf{b} = \begin{bmatrix} a_0 \\ a_1 \\ c_1 \\ \vdots \\ \vdots \\ \vdots \\ a_m \\ c_m \end{bmatrix}, \quad \mathbf{B} = \begin{bmatrix} g_1 \\ g_2 \\ \vdots \\ g_i \\ f_1 \\ f_2 \\ \vdots \\ f_j \end{bmatrix},$$

$$N^{a_v}_{j,v} = \sum_{v=1}^{m} [(v(\tfrac{1}{R})^v r_j^{v-1} \cos(v\theta)\cos\theta + (-v)(\tfrac{1}{R})^v r_j^v \sin(v\theta)\tfrac{-\sin\theta}{r_j})n_x \\ + (v(\tfrac{1}{R})^v r_j^{v-1} \cos(v\theta)\sin\theta + (-v)(\tfrac{1}{R})^v r_j^v \sin(v\theta)\tfrac{\cos\theta}{r_j})n_y] \tag{38}$$

$$N^{c_v}_{j,v} = \sum_{v=1}^{m} [(v(\tfrac{1}{R})^v r_j^{v-1} \sin(v\theta)\cos\theta + v(\tfrac{1}{R})^v r_j^v \cos(v\theta)\tfrac{-\sin\theta}{r_j})n_x \\ + (v(\tfrac{1}{R})^v r_j^{v-1} \sin(v\theta)\sin\theta + v(\tfrac{1}{R})^v r_j^v \cos(v\theta)\tfrac{\cos\theta}{r_j})n_y] \tag{39}$$

where $\mathbf{T}$ represents a $l \times M$ matrix, $M = 2m+1$, $\mathbf{b}$ represents a $M \times 1$ vector of unknown coefficients, $\mathbf{B}$ denotes a vector (size of $l \times 1$) of given functions at boundary points, $l$ represents the number of the boundary points and $M$ represents the terms of the Trefftz order. $i \leq l$ and $j \leq l$ in which $i$ and $j$ are the number of boundary points for Dirichlet and Neumann boundary conditions, respectively. $g_1$, $g_2$, ..., $g_i$ and $f_1$, $f_2$, ..., $f_j$ denote the boundary values for Dirichlet and Neumann boundary conditions, respectively.

In this article, we adopt the domain decomposition method (DDM) [37,38] to solve the nonlinear moving boundary problems in heterogeneous geological media. The DDM is commonly used to solve the problem with different physical characteristics in each subdomain. We first split the domain into two subdomains which are intersected only at the interface. Hence, each subdomain can be regarded as an independent soil layer with its own hydraulic conductivity. At the interface, the flux and the head must satisfy the continuity condition. For instance, we consider a rectangular domain, $\Omega$, which can be split into two intersected subdomains, $\Omega_1$ and $\Omega_2$. Figure 3 shows that the rectangular domain is divided into $\Gamma_1, \Gamma_2, \dots, \Gamma_8$ sub boundaries where $\Gamma_1, \Gamma_2, \dots, \Gamma_4$ and $\Gamma_5, \Gamma_6, \dots, \Gamma_8$ are sub boundaries of subdomains $\Omega_1$. and $\Omega_2$, respectively. At the interface, the sub boundaries, $\Gamma_2$ and $\Gamma_6$, are overlapped at the same location. Therefore, additional boundary conditions are imposed on the boundary points to ensure the flux and the head at the interface must be the same.

$$\varphi\big|_{\Gamma_2} = \varphi\big|_{\Gamma_6}, \ \frac{\partial\varphi}{\partial n}\Big|_{\Gamma_2} = \frac{\partial\varphi}{\partial n}\Big|_{\Gamma_6}. \tag{40}$$

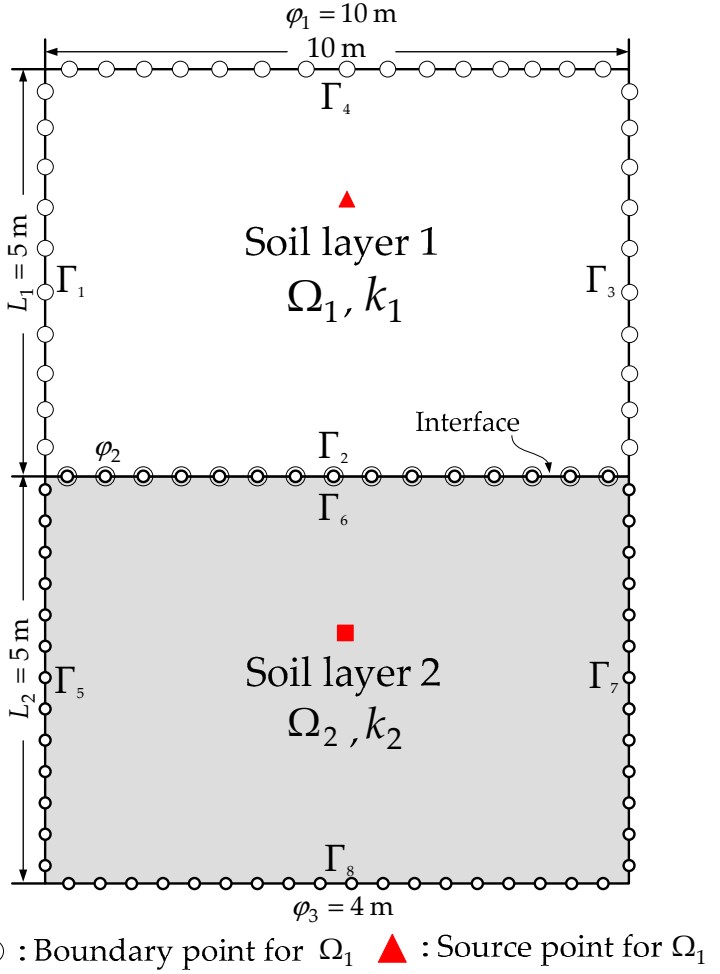

**Figure 3.** The cross section and the collocation scheme of the two layered soil for the analysis.

Matching all given boundary conditions, we may obtain a system of linear equations as

$$\mathbf{T}_D \mathbf{b}_D = \mathbf{B}_D, \tag{41}$$

$$\mathbf{T}_D = \begin{bmatrix} \mathbf{T}_{\Omega_1} & \mathbf{0}_{\Omega_2} \\ \mathbf{T}_I|_{\Gamma_2} & \mathbf{T}_I|_{\Gamma_6} \\ \mathbf{0}_{\Omega_1} & \mathbf{T}_{\Omega_2} \end{bmatrix}, \ \mathbf{b}_D = \begin{bmatrix} \mathbf{b}_{\Omega_1} \\ \mathbf{b}_{\Omega_2} \end{bmatrix}, \ \mathbf{B}_D = \begin{bmatrix} \mathbf{B}_{\Omega_1} \\ \mathbf{B}_I \\ \mathbf{B}_{\Omega_2} \end{bmatrix}, \tag{42}$$

where $\mathbf{T}_{\Omega_1}$ with the size of $l_1 \times M_1$ and $\mathbf{T}_{\Omega_2}$ with the size of $l_2 \times M_2$ are the **T** matrix shown in Equation (37) for $\Omega_1$ and $\Omega_2$, respectively. $l_1$ and $l_2$ are the number of boundary points; $M_1$ and $M_2$ are the T-complete basis function order for $\Omega_1$ and $\Omega_2$, respectively. $\mathbf{T}_I|_{\Gamma_2}$ of the boundary $\Gamma_2$ with the size of $l_I \times M_1$ and $\mathbf{T}_I|_{\Gamma_6}$ of the boundary $\Gamma_6$ with the size of $l_I \times M_2$ are matrices at the interface. $l_I$ represents the boundary point number at the interface, $\mathbf{0}_{\Omega_1}$ and $\mathbf{0}_{\Omega_2}$ are matrices which all values are zero with the size of $l_2 \times M_1$ and $l_1 \times M_2$, respectively. $\mathbf{b}_{\Omega_1}$ denotes a $M_1 \times 1$ vector of unknown coefficients of $\Omega_1$, $\mathbf{b}_{\Omega_2}$ denotes a $M_2 \times 1$ vector of unknown coefficients of $\Omega_2$. $\mathbf{B}_{\Omega_1}$ and $\mathbf{B}_{\Omega_2}$ denote vectors of given functions at boundary points of $\Omega_1$ and $\Omega_2$, respectively. $B_I = \begin{bmatrix} \mathbf{0}_g & \mathbf{0}_f \end{bmatrix}^T$, $\mathbf{0}_g$ and $\mathbf{0}_f$ are vectors which all values are zero with the size of $l_I \times 1$. The total head can be determined by collocating the inner points within subdomains, $\Omega_1$ and $\Omega_2$. Consequently, the value of the total head, $\varphi$, can then be approximated by using Equation (33).

### 3.3. The Iterative Scheme for Solving Free Surface

The nonlinearity of the moving surface flow is caused by the nonlinear boundary conditions. For solving free surface problems with nonlinear boundary conditions, the iterative scheme must be used in the numerical modeling. Due to the difficulty of the computation of the Jacobian matrix for Newton's method, the Picard scheme is adopted in this study. Applying boundary conditions, we obtain

$$\varphi(\mathbf{x}_k) \approx \sum_{v=1}^{m} \mathbf{b}_v \mathbf{T}_v(\mathbf{x}_k) = g(\mathbf{x}_k), \tag{43}$$

$$\frac{\partial \varphi(\mathbf{x}_k)}{\partial n} \approx \sum_{v=1}^{m} \mathbf{b}_v \frac{\partial}{\partial n} \mathbf{T}_v(\mathbf{x}_k) = f(\mathbf{x}_k), \tag{44}$$

where $k$ denotes the index of the collocation points on the free surface to be updated. The head at $J^{th}$ iteration is given as

$$\varphi^J(\mathbf{x}_k) \approx \sum_{v=1}^{m} \mathbf{b}_v^J \mathbf{T}_v(\mathbf{x}_k^J), \tag{45}$$

$$\frac{\partial \varphi^J(\mathbf{x}_k)}{\partial n} = \nabla\varphi^J \cdot \vec{n} \approx \sum_{v=1}^{m} \mathbf{b}_v^J \frac{\partial}{\partial n} \mathbf{T}_v(\mathbf{x}_k^J), \tag{46}$$

where $J$ denotes the number of iteration steps. We may obtain the following iterative equation [39].

$$\varphi^J(\mathbf{x}_k) = \varphi^{J-1}(\mathbf{x}_k) + \beta(\varphi^J(\mathbf{x}_k) - \varphi^{J-1}(\mathbf{x}_k)), \tag{47}$$

where $\beta$ is the under–relaxation factor and $\varphi^J(\mathbf{x}_k)$ is the head to be updated. The value of $\beta$ is ranging from 0 to 1. The iterative process begins from the first guess values for the moving surface and ceases while the stopping condition expressed in the following equation is achieved.

$$\varepsilon = \frac{\sqrt{\sum_{k=1}^{ni} (\varphi^J(\mathbf{x}_k) - \varphi^{J-1}(\mathbf{x}_k))^2}}{\sqrt{\sum_{k=1}^{ni} (\varphi^{J-1}(\mathbf{x}_k))^2}} \leq 10^{-4}, \tag{48}$$

where $ni$ is the collocation point number on the free surface.

## 4. Validation Examples

### 4.1. Laminar Flow around a Cylinder

The first example under consideration is a laminar flow around a cylinder. The dimensions of the problem are depicted in Figure 4a. The radius of the cylinder at the center is 1 m. Because the geometry of the problem is clearly symmetric, we considered the half symmetry model. For a two-dimensional domain, $\Omega$, enclosed by a boundary, the Laplace equation is expressed as

$$\Delta\varphi = 0 \text{ in } \Omega \tag{49}$$

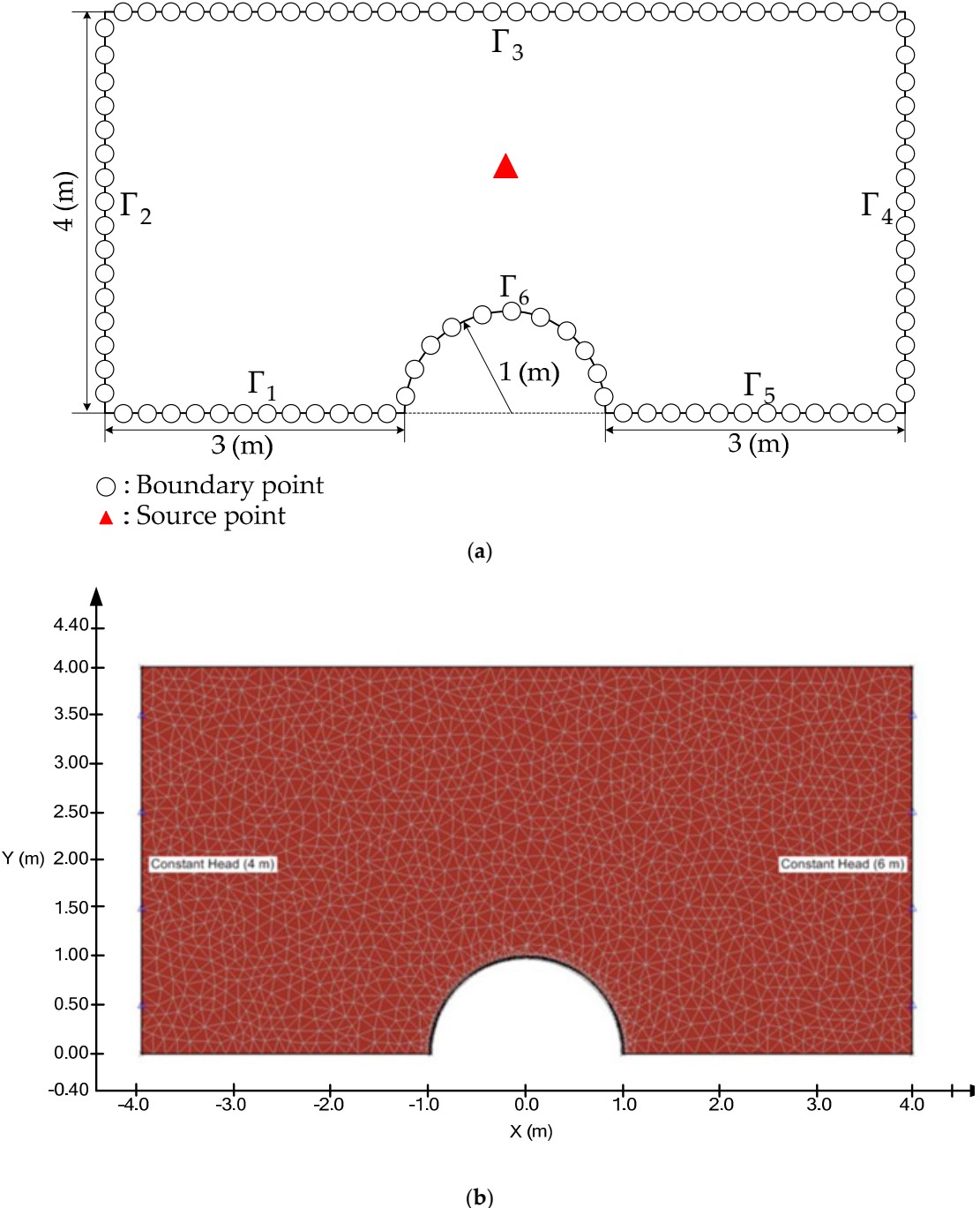

**Figure 4.** Comparison of the domain discretization. (**a**) The CTM and (**b**) the finite element method.

The Dirichlet and Neumann data are applied on the domain boundary, $\Gamma$, including $\Gamma_1, \Gamma_2, \ldots, \Gamma_6$, as shown in Figure 4a. On $\Gamma_2$ and $\Gamma_4$, the Dirichlet data are $\phi = 4$ m on $\Gamma_2$ and $\varphi = 6$ m on $\Gamma_4$. On $\Gamma_1$, $\Gamma_3$, $\Gamma_5$ and $\Gamma_6$, the no–flow Neumann boundary data are given as follows.

$$\frac{\partial \varphi}{\partial n} = 0. \tag{50}$$

The order of the Trefftz basis functions, $m$, is 150. Totally, 900 collocation points including boundary points and sources are uniformly placed on the domain boundary, as shown in Figure 4a. In order to examine accuracy of the proposed method, 7786 inner points are collocated within the domain

boundary. The computed results are validated with SVFLUX [40] which is a finite element seepage analysis program. Figure 4b shows the finite element mesh of SVFLUX. The results of the computed head are compared with the exact solution, as shown in Figure 5. It is found that the numerical solutions agree very well with those of the SVFLUX.

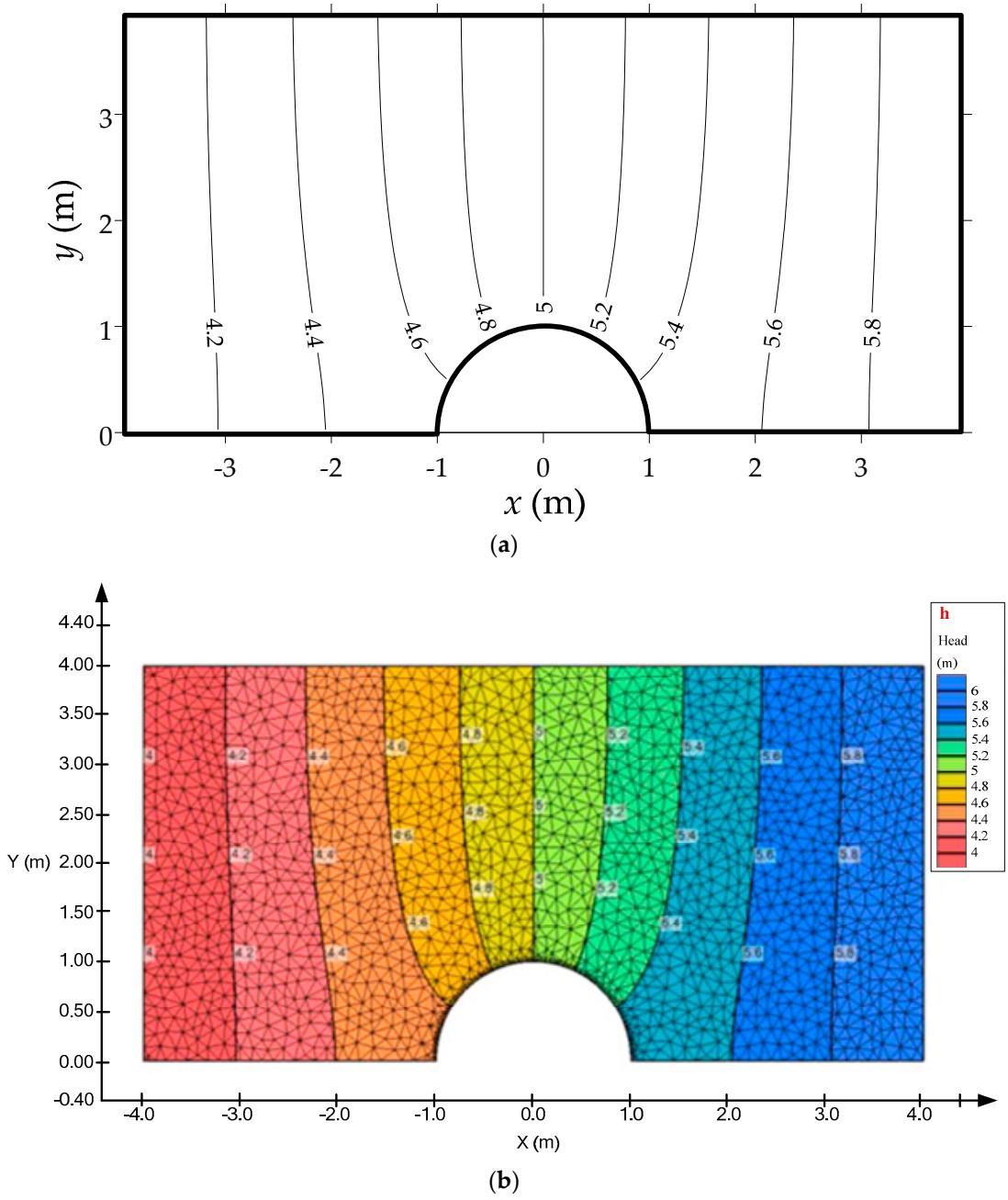

**Figure 5.** Comparison of computed results of the CTM with those from SVFLUX. (**a**) Computed results of the CTM and (**b**) SVFLUX results.

### 4.2. Nonlinear Moving Surface through a Rectangular Dam

The second example is a nonlinear moving surface through a rectangular dam, as shown in Figure 1a. This problem can be viewed as a benchmark problem in geotechnical engineering for finding the position of the moving boundary [3,11,14]. The upstream and downstream heads are 24 m and 4 m, respectively. The dimensions of the problem are depicted in Figure 6. This rectangular dam is assembled with five boundary lines, including $\Gamma_1$, $\Gamma_2$, $\Gamma_3$, $\Gamma_4$ and $\Gamma_5$, as depicted in Figure 1b.

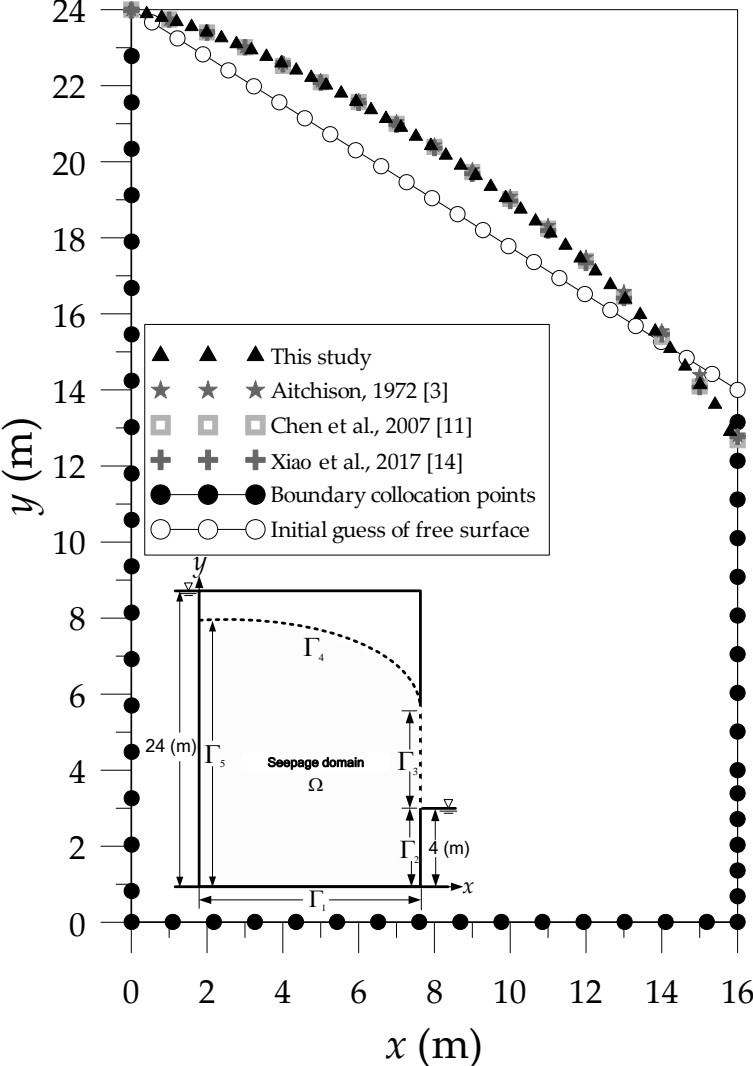

**Figure 6.** Comparison of moving boundary results with other numerical methods.

The order of the Trefftz basis functions, *m*, is 75. We collocate 120 collocation points on moving boundary, as depicted in Figure 6. Since the process for finding the position of the nonlinear free surface is regarded as an inverse problem, the location of the separation point may also need to be examined. In this study, the initial guess of the separation point is at 14.2 m.

The numerical solutions of the free surface are shown in Figure 6 and the results are compared with those from other methods, such as Aitchison [3], Chen et al. [11] and Xiao et al. [14]. Figure 7 shows that for solving the free surface the number of iteration step is 112 to reach the stopping criterion using the proposed iteration scheme. To further explore the accuracy of the computed results, we compare the computed location of the separation point with those from other numerical methods [3,11,14]. As depicted in Table 1, the position of the free surface is almost identical with other methods. The results of the separation point calculated by three different methods [3,11,14] are 12.79, 12.68 and 12.84 m, respectively. It is found that the location of the separation point by using the CTM is 12.85 m which is close to that from previous studies.

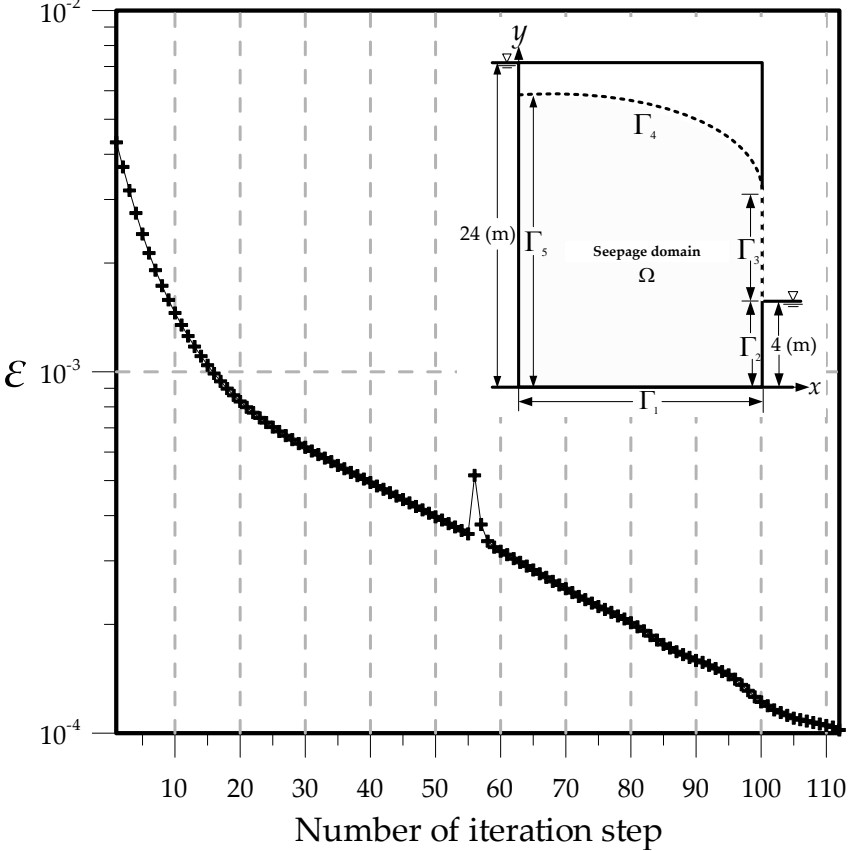

**Figure 7.** The convergence of the iteration step for finding the free surface.

**Table 1.** Comparison of computed result of the separation point with those from references.

| Reference | Height (m) |
| --- | --- |
| This study | 12.85 |
| Aitchison [3] | 12.79 |
| Chen, Hsiao, Chiu and Lee [11] | 12.68 |
| Xiao, Ku, Liu, Fan and Yeih [14] | 12.84 |

*4.3. Nonlinear Moving Surface through an Earth Dam*

The third example is a nonlinear moving surface through an earth dam, as shown in Figure 8. The upstream and downstream hear are 18 m and 8 m, respectively. The dimensions of the problem are depicted in Figure 9a. The boundaries of the earth dam include $\Gamma_1$, $\Gamma_2$, $\Gamma_3$, $\Gamma_4$ and $\Gamma_5$ in which $\Gamma_2$ and $\Gamma_5$ are the downstream and upstream boundaries, as depicted in Figure 8. The boundary values are given as

$$H_1 = 18 \text{ m on } \Gamma_5 \text{ and } H_2 = 8 \text{ m on } \Gamma_2. \tag{51}$$

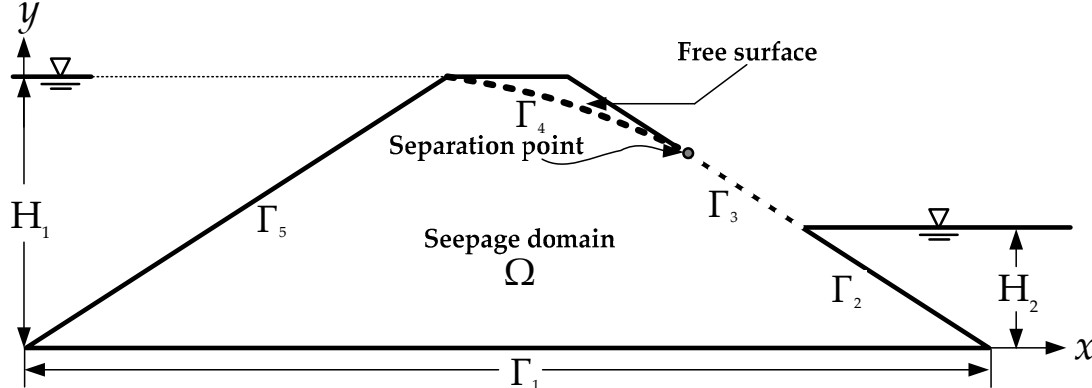

**Figure 8.** The cross section of the homogeneous earth dam for the analysis.

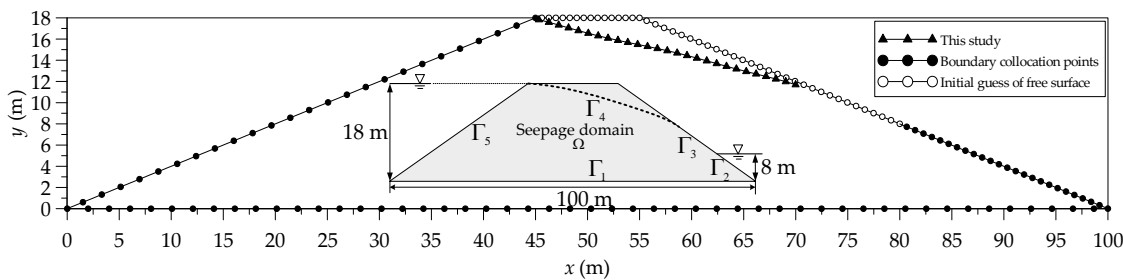

**(a)**

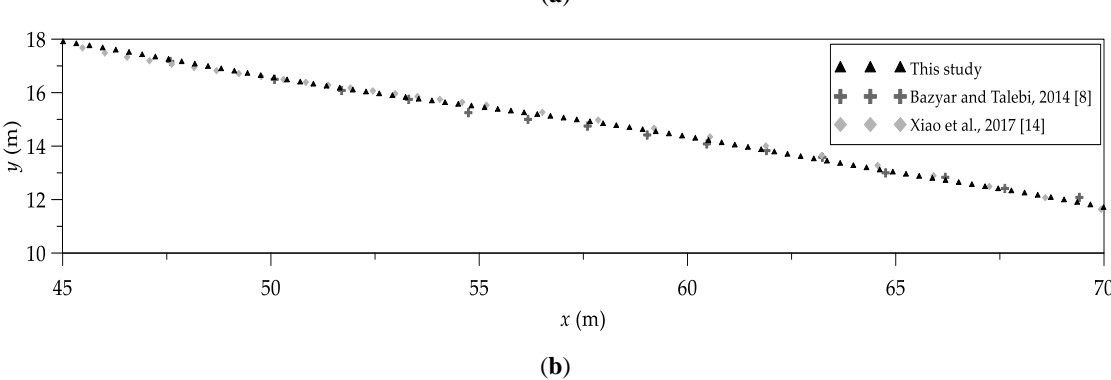

**(b)**

**Figure 9.** Free surface flow through a homogeneous dam. (**a**) Initial guess and the computed free surface and (**b**) comparison of the results with those from other methods.

$\Gamma_1$ is the bottom of the dam where the no–flow condition is given as

$$\frac{\partial \varphi}{\partial n} = 0 \text{ on } \Gamma_1. \tag{52}$$

On $\Gamma_3$, the seepage face boundary condition is as follows.

$$\varphi = Y(x) \text{ on } \Gamma_3. \tag{53}$$

On $\Gamma_4$, the over–specified moving boundary conditions are as follows.

$$\frac{\partial \varphi}{\partial n} = 0, \ \varphi = Y(x) \text{ on } \Gamma_4. \tag{54}$$

For this example, the Trefftz basis function order, *m*, is 150. 750 points are collocated on the boundary. The first guess of the free surface is depicted in Figure 9a. The numerical solutions of the free surface are compared with those from other methods [8,14], as shown in Figure 9b. The results illustrate that the computed results are almost identical with other methods.

### 4.4. Flow through Two Layered Soils

The modeling of two-dimensional heterogeneous isotropic subsurface flow in two layered soils is depicted in Figure 3. The coefficients of the permeability with extreme contrasts for two different soils, $k_1$ and $k_2$, are adopted in which the $k_1 = 10^{-1}$ and $k_2 = 10^{-15}$ cm/s. The analytical solution of this example as shown in follows.

$$\varphi = \frac{\varphi_2 - \varphi_1}{L_1} x + \varphi_1, \ 0 \le x \le L_1, \tag{55}$$

$$\varphi = \frac{\varphi_3 - \varphi_2}{L_2} x + \left(\varphi_2 - \frac{\varphi_3 - \varphi_2}{L_2} L_1\right), \ L_1 \le x \le L_1 + L_2, \tag{56}$$

$$\varphi_2 = \frac{k_1 \varphi_1 L_2 + k_2 \varphi_3 L_1}{k_1 L_2 + k_2 L_1}, \tag{57}$$

where $L_1$ is the width of the layer 1, $L_2$ is the width of the layer 2 and $\varphi_2$ is the head at the interface. The domain is split into two sub-domains which are simply connected. For each sub-domain, 112 boundary points are uniformly collocated. Figure 3 depicts that the rectangular domain boundary is split into eight sub–boundaries: $\Gamma_1, \Gamma_2, \dots, \Gamma_8$. At the interface, we have the following additional boundary conditions.

$$\varphi|_{\Gamma_2} = \varphi|_{\Gamma_6}, \ \frac{\partial \varphi}{\partial n}\bigg|_{\Gamma_2} = \frac{\partial \varphi}{\partial n}\bigg|_{\Gamma_6}. \tag{58}$$

Totally, 1520 interior points are collocated within the domain to approximate the head for examining the accuracy. The results of the computed head are compared with the exact solution, as shown in Figure 10. In addition, the accuracy of the results can reach to the order of $10^{-11}$, as shown in Figure 11.

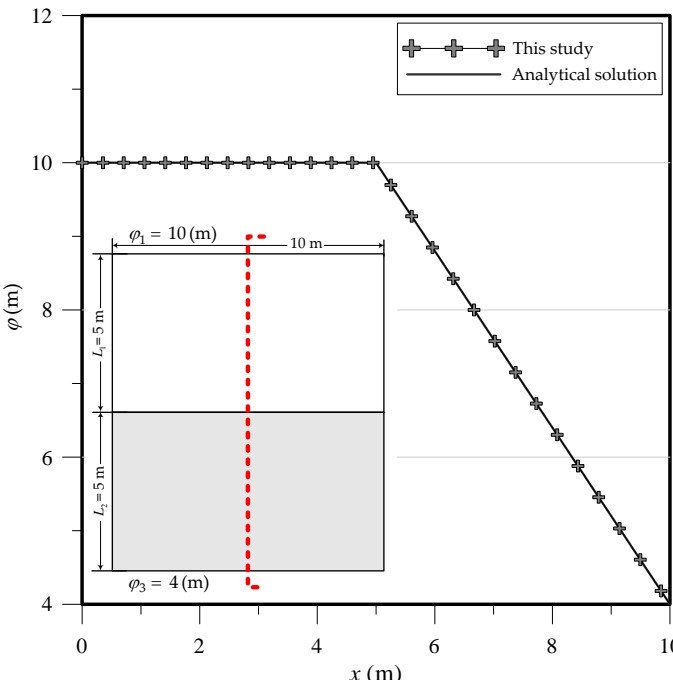

**Figure 10.** The computed head and the analytical solution in two-layered soils.

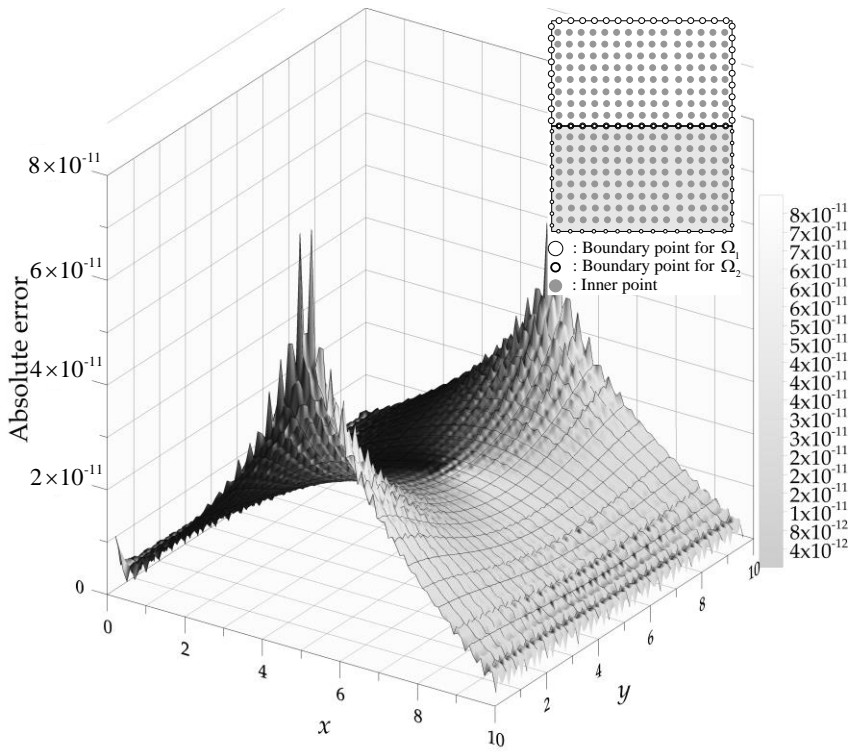

**Figure 11.** Absolute error of the example in two-layered soils.

### 4.5. Nonlinear Moving Surface through a Zoned–Earth Dam

The last case is a nonlinear moving surface through a zoned–earth dam, as shown in Figure 12. For the zoned–earth dam, the upstream and downstream heads are 18 m and 2 m, respectively. The dimensions of the problem are depicted in Figure 12. The values of the permeability for the upstream shell, the core and the downstream shell are $1.43 \times 10^{-4}$, $1.43 \times 10^{-5}$ and $1.43 \times 10^{-4}$ cm/s, respectively.

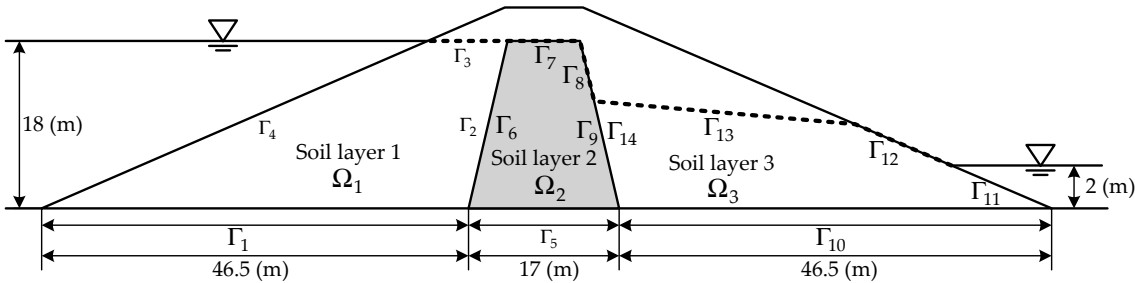

**Figure 12.** The cross section and soil layer configuration of the zoned–earth dam for the analysis.

For the zoned–earth dam, we divide the domain into three smaller sub-regions, as shown in Figure 13. For each sub-region, we collocate 400, 216 and 191 points on the sub-boundaries for the first, second and third sub-regions, respectively. Besides, we place 50, 66 and 66 collocation points on the moving boundaries, respectively. To validate the results, we compare the computed free surface with that from the finite difference seepage analysis program SEEP2D [41], as shown in Figure 14. It is found that the numerical results agree well with those obtained from the SEEP2D.

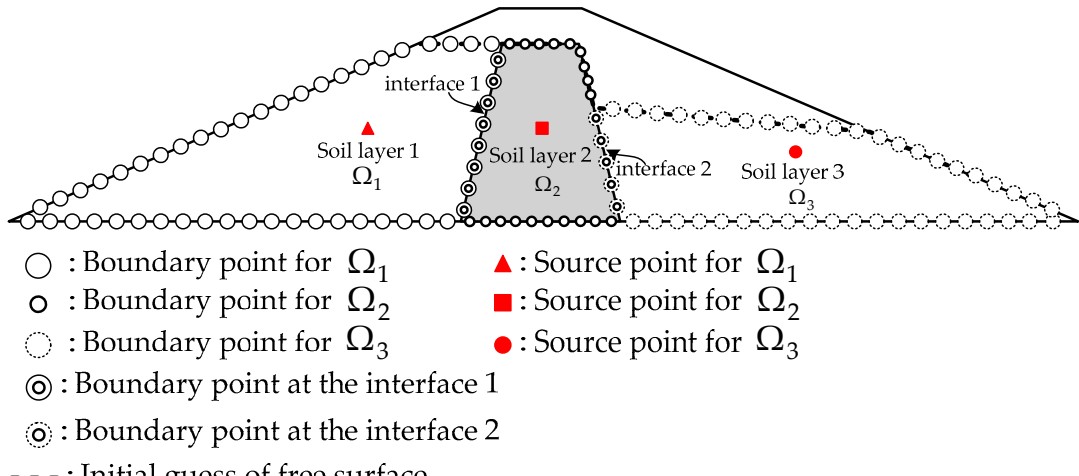

**Figure 13.** The collocation scheme of the zoned-earth dam for the DDM.

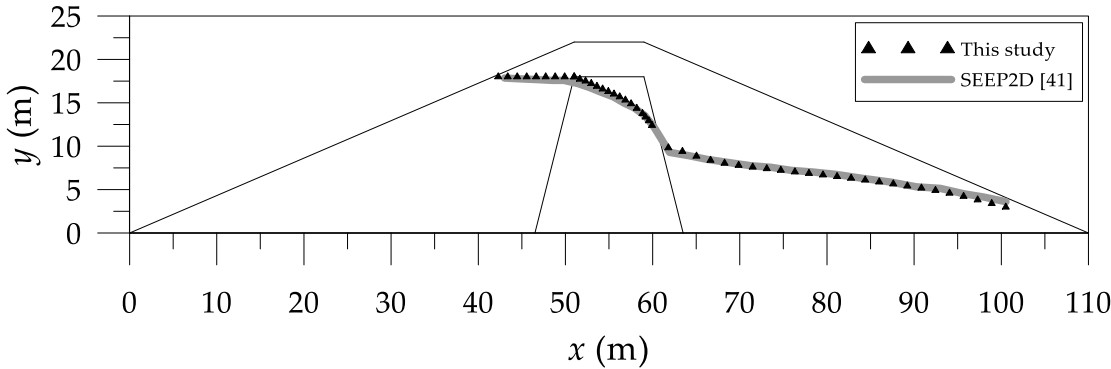

**Figure 14.** Comparison of the computed free surface with SEEP2D.

## 5. Discussion

In this article, the CTM is adopted to solve the nonlinear moving boundary problems in layered heterogeneous media. Because of the characteristics of the non-linearity, solving nonlinear moving boundary problems with a moving surface remains a challenge. For modeling moving surface problems with nonlinear boundary conditions, the iterative scheme must be used. The sophisticated automatic mesh generation may be required using conventional mesh-based approaches. In addition, the complicated remeshing process in the iterative scheme may lead to problems of the convergence. In this study, we just need to place the boundary points on the domain boundary. Furthermore, only boundary nodes are required to adjust during the iteration process for find the moving boundary. Comparing with the traditional numerical methods, the proposed method is highly accurate. Therefore, the proposed method is advantageous for the nonlinear moving boundary analysis with a phreatic line.

To solve the flow through the layered soils, we adopt the CTM in conjunction with the DDM so that the hydraulic conductivity in each subdomain can be different. To verify the proposed method, numerical examples with nonlinear moving boundary are conducted. Besides, free surface flow through a zoned–earth dam is also carried out. The comparison of the results using the DDM with the exact solution depicts that the CTM with the use of the DDM can reach the accuracy in the order of $10^{-11}$. Although the CTM may be regarded as an attractive boundary–type meshless method, limitations for the applications may still remain due to the ill-posedness of the method.

## 6. Conclusions

This paper presents a study on solving nonlinear moving boundary problems in heterogeneous soils using the CTM. The method is verified by several numerical examples. Application examples are also carried out. The findings are concluded.

The appearance of heterogeneous soils is often found in free surface flow problems. In the past, the CTM is usually limited to linear, homogeneous problems. In this article, we propose a novel idea for solving nonlinear moving boundary problems in layered heterogeneous soils using the collocation meshless method. The results show that the proposed method can be used to deal with nonlinear moving boundary problems in heterogeneity with large permeability contrasts. The method proposed in this study is capable of solving nonlinear moving boundary problems in layered heterogeneous media. However, it is still limited to the layered or zoned porous media in which the porous medium is still homogenous in each zoned domain. In addition, the proposed method can only be applied in saturated soils.

**Author Contributions:** Conceptualization, C.-Y.K.; Data curation, C.-Y.L.; Methodology, J.-E.X.; Validation, J.-E.X.; Writing—original draft, C.-Y.K.; Writing—review & editing, C.-Y.K. and J.-E.X.

**Funding:** This research was funded by Ministry of Science and Technology of the Republic of China under grant MOST 108-2119-M-019-001.

**Acknowledgments:** The authors thank the Ministry of Science and Technology for the generous support. The first author would like to express his gratitude to former student, Yi-Wun Chen, for her assistance of this study.

**Conflicts of Interest:** The authors declare that there are no conflicts of interest regarding the publication of this article.

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
