# Peer review of "On Solving Nonlinear Moving Boundary Problems with Heterogeneity Using the Collocation Meshless Method"

_water, doi:10.3390/w11040835_

Round 1

Reviewer 1 Report

I have now read this manuscript. It is a computational work that deals with a numerical method for solving the Laplace equation with one moving boundary. This appears to have applications in seepage flow problems. The authors apply the Trefftz method for obtaining the solution, this having the computational advantage that avoids generation of mesh structure needed by standard finite element method (FEM) for solving this kind of problem.

The authors obtain their solutions to different problems of the same kind with an accuracy very similar to that of FEM. Taking into account that their method is much simpler, I think it justifies the relevance of their results within their research field, which may be a bit too specialized.

In any case, the manuscript is correctly presented and structured, quality of writing is ok (can be improved though) and for those reasons I recommend the paper for publication in the Water journal.

My only suggestion is perhaps to follow the same order for panels in Figs. 4 and 5; i.e. Their method results should correspond to the same panel (either a or b in both figures) in order to avoid confusion to the reader.

Author Response

The responses of author to the reviewer 's comments are as the follows.

Reviewer 1

My only suggestion is perhaps to follow the same order for panels in Figs. 4 and 5; i.e. Their method results should correspond to the same panel (either a or b in both figures) in order to avoid confusion to the reader.

Response:

Thank you so much for this invaluable comment. Figs. 4 and 5 have been revised with the same order, as shown in the page 10 and page 11.

Reviewer 2 Report

Manuscript is potentially publishable after the following clarifications and manus improvements:

1) Clearly state the limitations of this methodology because it is also not able to consider real heterogeneous porous media (except zonal homogeneous domainsor infiltration in unsaturated zone, application of globally supported basis functions, ill-posedness ....

2) Dependence of the number of decomposition domains and needed iterations (inner in linear/non-linear solvers  and outer - number of global iterations over all subdomains)

This dependence implies that procedure is influenced by number of domains. Illustrate this issue by corresponding analysis and/or discussion.

3) Equation 32. Please, explain the role of characteristic length and why it reduces ill-posedness of the resulting system of equations.

4) Generally, all verification examples have lack of the presented the convergence analysis. Namely, it is needed to show dependence of number of collocation points on method accuracy as well as relationship to the for instance the classic FEM. 

Author Response

The responses of reviewer's comments are as the attached file.

This manuscript is a resubmission of an earlier submission. The following is a list of the peer review reports and author responses from that submission.